# Snake Venom C-Type Lectin-like Protein *Vaa*-Snaclec-3/2 Efficiently Prevents Carotid Artery Thrombosis in a Mouse Model Without Compromising Blood Coagulation

**DOI:** 10.3390/toxins17110523

**Published:** 2025-10-23

**Authors:** Monika C. Žužek, Igor Križaj, Miran Brvar, Tomaž Trobec, Simona Kranjc Brezar, Mojca Dobaja Borak, Adrijana Leonardi, Kity Požek, Milka Vrecl, Robert Frangež

**Affiliations:** 1Institute of Preclinical Sciences, Veterinary Faculty, University of Ljubljana, 1000 Ljubljana, Slovenia; monikacecilija.zuzek@vf.uni-lj.si (M.C.Ž.); tomaz.trobec@vf.uni-lj.si (T.T.); milka.vreclfazarinc@vf.uni-lj.si (M.V.); 2Department of Molecular and Biomedical Sciences, Jožef Stefan Institute, 1000 Ljubljana, Slovenia; igor.krizaj@ijs.si (I.K.); adrijana.leonardi@ijs.si (A.L.); kity.pozek@ijs.si (K.P.); 3Centre for Clinical Toxicology and Pharmacology, University Medical Centre Ljubljana, 1000 Ljubljana, Slovenia; miran.brvar@kclj.si (M.B.); mojca.dobaja@kclj.si (M.D.B.); 4Center for Clinical Physiology, Faculty of Medicine, University of Ljubljana, 1000 Ljubljana, Slovenia; 5Department of Experimental Oncology, Institute of Oncology Ljubljana, 1000 Ljubljana, Slovenia; skranjc@onko-i.si; 6Doctoral School, Faculty of Medicine, University of Ljubljana, 1000 Ljubljana, Slovenia

**Keywords:** snake venom, antithrombotic, snaclec, arterial thrombosis, mice

## Abstract

Platelets play pivotal roles in thromboembolic diseases, such as myocardial infarction and ischemic stroke. In patients envenomed by the snake *Vipera a. ammodytes* (*Vaa*), pronounced and transient thrombocytopenia without bleeding is observed. We previously showed that *Vaa*-snaclec-3/2, the snake venom C-type lectin-like protein, mediates this effect ex vivo. Here, we extended our study of the antithrombotic potential of this protein *in vivo* using a mouse model of ferric chloride (FeCl_3_)-induced carotid artery thrombosis. Prior to inducing thrombus formation, the mice received 1, 5, 10, 20, or 50 μg/kg *Vaa*-snaclec-3/2 intravenously. Afterward, the arterial blood flow was monitored with a perivascular Doppler probe. Additionally, the platelet count in the peripheral venous blood; tail bleeding time; and liver, lung, kidney, spleen, and heart histology were evaluated. The lowest dose of *Vaa*-snaclec-3/2 that we showed to cause severe thrombocytopenia and completely inhibit FeCl_3_-induced thrombus formation was 20 µg/kg. This dose prolonged the median tail bleeding time from 86.5 to 153.5 s but did not induce acute spontaneous hemorrhage, as demonstrated by histological analysis. Histology revealed no signs of apoptosis, necrosis or other degenerative changes in the inspected organs of mice exposed to 20 μg/kg *Vaa*-snaclec-3/2. Platelet clusters were observed only in the lungs, which appear to be the primary site of platelet sequestration and the cause of thrombocytopenia. Taken together, our findings highlight the high potential of *Vaa*-snaclec-3/2 as a safe and effective antithrombotic agent for the transient prevention of thrombosis in acute clinical settings.

## 1. Introduction

Cardiovascular diseases (CVDs) are the leading cause of death worldwide and are responsible for 32% of all deaths. Thrombosis-related CVDs (heart attack and stroke) are the leading cause of death from CVDs, accounting for 85% of CVD-related deaths [1]. Snake venoms are rich sources of components with antithrombotic activity, and these components have inspired the development of numerous therapeutic anticoagulants [2]. *Vipera b. berus* (*Vbb*) and *Vipera a. ammodytes* (*Vaa*) are the only important venomous snakes in Slovenia, and some components of their venoms are of medical importance. In humans who are bitten by *Vaa*, pronounced and transient thrombocytopenia without bleeding is observed [3]. Once this *Vaa* venom-induced severe thrombocytopenia is reversed, the rotational thromboelastometry and aggregometry results, which provide information about the overall kinetics of hemostasis and platelet function, are within the normal range. In addition, the physiological function of platelets is fully preserved [4]. Various snake venom components can cause thrombocytopenia via different mechanisms. These components include nonenzymatic, direct-acting toxins that belong to the snake venom C-type lectin-like protein (snaclec) family [5,6].

Snaclecs are found primarily in snakes of the Viperidae family. Structurally, they are heterodimers composed of α and β subunits, each featuring a C-type lectin-like domain, which adopts a conserved fold stabilised by disulphide bridges but lacks the classical carbohydrate- or calcium-binding activity seen in true C-type lectins [5]. Long loop pole-swapping between α- and β-subunits forms a concave interaction surface for binding different target molecules. These heterodimers can oligomerise into non-covalently or covalently bound complexes of two, three, four, or eight heterodimer units.

Snaclecs exert effects on the haemostatic system by binding to specific targets in this system, such as platelet receptors (e.g., GPIb, GPVI, α_2_β_1,_ CLEC-2), von Willebrand factor (vWF), or coagulation factors (e.g., factor IX and X) [5,7,8,9]. The latter interact with snaclecs via the Gla domains in a calcium-dependent manner. It has been observed that oligomeric snaclecs are much more effective at clustering receptors and activating platelets than their αβ heterodimeric variants [5]. Depending on their target, snaclecs can either activate or inhibit platelet function and interfere with the coagulation cascade, leading to diverse clinical effects such as thrombosis, bleeding, or thrombocytopenia in envenomed victims [10,11,12]. Their high specificity and affinity for components of the human haemostatic system have made them valuable tools in research and promising leads for developing novel antiplatelet or anticoagulant therapies [13,14,15].

Recently, *Vaa*-snaclec-3/2 was isolated from *Vaa* venom. It is a non-glycosylated, covalent heterodimeric protein composed of *Vaa*-snaclec-3 (A0A1I9KNN1_VIPAA) as the α-subunit and *Vaa*-snaclec-2 (A0A1I9KNS2_VIPAA) as the β-subunit [16]. It has been shown to form (αβ)_2_ oligomers as well. This protein agglutinates platelets by binding to the platelet membrane GPIb-IX-V receptor complex (GPIb), causing thrombocytopenia [17].

GPIb is the major adhesion receptor that mediates the shear-dependent adhesion of platelets to vWF in the first stage of hemostasis [18]. The binding of vWF to GPIb is a promising target for the development of new therapies for arterial and venous thrombosis [19]. Hemostasis and occlusive thrombus formation remain largely unaffected in mice with various degrees of platelet count reduction [20]. Our preliminary results revealed that 50 µg/kg *Vaa*-snaclec-3/2 induces severe thrombocytopenia by reducing platelet counts by up to 98% and protects mice from ferric chloride (FeCl_3_)-induced carotid artery thrombosis without causing bleeding [17]. In humans, thrombocytopenia induced by *Vaa* venom was reversed within 1 h by the administration of Fab fragments of immunoglobulin G (IgG) antibodies produced using the entire venom [4]. Since *Vaa*-snaclec-3/2 has antiplatelet properties, it is considered a promising antithrombotic drug candidate. Therefore, the main objective of this work was to evaluate its antithrombotic efficacy and safety *in vivo* using a mouse model of vascular injury.

## 2. Results

### 2.1. Antithrombotic Effects of Vaa-Snaclec-3/2 In Vivo

The antithrombotic activity of *Vaa*-snaclec-3/2 was investigated in a mouse model of FeCl_3_-induced carotid artery thrombosis [9]. Doses of 1, 5, 10, 20 and 50 μg/kg *Vaa*-snaclec-3/2 (n = 5 per dose) were administered intravenously (*i.v.*) 5 min prior to the induction of thrombus formation by FeCl_3_. *Vaa*-snaclec-3/2 had a dose-dependent protective effect against thrombus formation. No protection was observed at the lowest dose (1 μg/kg). Partial protection was observed at 5 μg/kg and 10 μg/kg *Vaa*-snaclec-3/2, preventing carotid obstruction in 2 of 5 mice (40%) in each group. However, mice treated with 20 μg/kg *Vaa*-snaclec-3/2 or 50 μg/kg *Vaa*-snaclec-3/2 were all resistant to FeCl_3_-induced carotid artery thrombosis, as was also observed in mice treated with heparin alone (positive control, n = 5) (Figure 1). In contrast, in the control group (n = 5), which was injected with 0.9% sterile saline solution only, FeCl_3_-induced thrombus formation resulted in complete arterial occlusion in all the experimental animals (Figure 1). The minimal dose of *Vaa*-snaclec-3/2 that completely prevented occlusion of the carotid artery can therefore be estimated to be between 10 and 20 μg/kg. *Vaa*-snaclec-3/2 is clearly a protein with strong antithrombotic activity in mice.

### 2.2. Effect of Vaa-Snaclec-3/2 on the Platelet Count

To explore the mechanism underlying the antithrombotic effect of *Vaa*-snaclec-3/2 in mice, we determined blood platelet counts in *Vaa*-snaclec-3/2-treated and untreated animals and compared the values. After the mouse blood flow measurements were completed, blood samples were collected from the retro-orbital sinuses of the animals, and platelet counts were determined. A maximal decrease in the platelet count (∼98%) was observed in the group that was treated with the highest dose of *Vaa*-snaclec-3/2 (50 μg/kg).

In this group, the platelet count decreased from 962 ± 61 × 10^9^/L to 19 ± 4 × 10^9^/L, which is less than 2% of the value recorded in the group of mice that were treated with sterile saline solution only (*p* = 0.001) [9]. In mice injected with 20 μg/kg *Vaa*-snaclec-3/2, the platelet count decreased by approximately 90% compared with that in the control group, from 962 ± 33 × 10^9^/L to 95 ± 15 × 10^9^/L (*p* = 0.001). In contrast, the platelet count in the heparin- and FeCl_3_-induced thrombus formation groups was comparable to that of the control group (Figure 2). *In vivo* experiments clearly revealed that *Vaa*-snaclec-3/2 inhibits thrombus formation by inducing profound thrombocytopenia.

To gain insight into the kinetics of thrombocytopenia in mice due to *Vaa*-snaclec-3/2, the platelet count was also determined 5 min after the injection of 20 µg/kg of the substance. If the platelet count decreased from 962 ± 33 × 10^9^/L to 95 ± 15 × 10^9^/L (n = 5) within 30 min, i.e., to 10% of the initial value, the major platelet count reduction occurred during the first 5 min, when it decreased to 18% of the initial value (to 171 ± 30 × 10^9^/L; n = 3). *Vaa*-snaclec-3/2 clearly causes thrombocytopenia very rapidly.

Injection of the antivenom VIPERFAV^®^ into mice 10 min after the injection of 20 µg/kg *Vaa*-snaclec-3/2 resulted in a partial recovery of the platelet count to 168 ± 18 × 10^9^/L (n = 7) 3 h later. The process of *Vaa*-snaclec-3/2-induced thrombocytopenia can thus be reversed by specific antivenom in mice.

### 2.3. Effect of Vaa-Snaclec-3/2 on Hemostasis

The *Vaa*-snaclec-3/2 at a dose of 20 μg/kg significantly reduced the number of platelets in the peripheral venous blood of mice (Figure 2) and completely prevented FeCl_3_-induced occlusion of the carotid artery (Figure 1). To determine the extent to which this dose of *Vaa*-snaclec-3/2 interferes with hemostasis, the tail bleeding time was measured following a bolus injection (*i.v.*) of 20 μg/kg *Vaa*-snaclec-3/2.

The median tail bleeding time of the mice in the control group was 86.5 s (IQR: 73.5 s to 144 s). However, 20 μg/kg *Vaa*-snaclec-3/2 prolonged the median tail bleeding time to 153.5 s (IQR: 132.5 s to 210 s), which was significantly longer than the value obtained in the control group (*p* = 0.028) (Figure 3). The reported normal bleeding times of different strains of mice vary between 66 and 288 s [21]. Therefore, the bleeding times of *Vaa*-snaclec-3/2-treated mice remained within the reference range, indicating that *Vaa*-snaclec-3/2 did not compromise hemostasis despite the 95% decrease in the platelet count, from 962 ± 61 × 10^9^/L to 42 ± 7 × 10^9^/L. The remaining platelets were clearly sufficient to maintain functional hemostasis.

### 2.4. Effect of Vaa-Snaclec-3/2 on Hemoglobin Concentration and Hematocrit

Intravenous injection of *Vaa*-snaclec-3/2 (20 or 50 μg/kg) did not significantly affect hemoglobin concentration or hematocrit values in peripheral venous blood compared with the control groups (*p* > 0.05). All experimental groups, including those treated with heparin or FeCl_3_ alone, showed comparable values (Figure 4).

### 2.5. Analysis of Mouse Organs Exposed to Vaa-Snaclec-3/2

#### 2.5.1. Effects of Vaa-Snaclec-3/2 on the Relative Mass of Mouse Organs

Changes in relative mass are considered sensitive indicators of chemically induced effects on organs, and these changes are typically assessed to determine drug-induced toxicity *in vivo* [22]. The effects of 20 μg/kg *Vaa*-snaclec-3/2 on the relative masses of selected organs are summarized in Table 1.

There were no statistically significant differences in the relative organ mass between the control and experimental groups (*p* > 0.05). Even when *Vaa*-snaclec-3/2 was administered at the highest dose (50 μg/kg), the relative organ mass did not significantly differ between the control group and the treated group (*p* > 0.05). We can thus conclude that *Vaa*-snaclec-3/2 did not induce acute effects, such as edema or hemorrhage, in the analyzed mouse organs.

#### 2.5.2. Histological Findings in the Vaa-Snaclec-3/2-Treated Mice

Representative histological sections of livers, kidneys, lungs and left ventricles from control mice and mice treated *i.v.* with *Vaa*-snaclec-3/2 (20 μg/kg) are shown in Figure 5. No obvious histological abnormalities were observed in the organ sections from the control mice (Figure 5, left panels). Organ sections from the *Vaa*-snaclec-3/2-treated mice retained their overall architecture (Figure 5, right panels). Liver sections from control mice showed preserved hepatocytes arranged in the laminae hepatis, as well as clearly visible central veins and sinusoids. Age-related karyomegaly was observed [19], which was not related to the treatment, as polyploidization was observed in both the control and treatment groups. In the *Vaa*-snaclec-3/2-treated mice, the hepatocytes appeared to be enlarged and exhibited distinct cell borders, and the sinusoids were indistinct (Figure 5, top right panel). Individual platelet agglutinates were, however, observed in the pulmonary veins. Additionally, brown-stained erythrocytes were found in the vasa recta venules near the corticomedullary junction of the kidney, as well as in the lungs and the red pulp of the spleen (Figure 5). These changes may reflect oxidative changes in hemoglobin rather than iron accumulation [20]. The histological changes observed are likely the result of adaptive processes as a metabolic response to the injected protein. No signs of apoptosis, necrosis or other degenerative changes were observed in the tissues of the organs harvested from mice exposed to a dose of 20 μg/kg *Vaa*-snaclec-3/2.

## 3. Discussion

Our *in vivo* study revealed that *Vaa*-snaclec-3/2 can prevent thrombus formation and protect mice against carotid artery occlusion; this was demonstrated by the measurement of vascular flow using a perivascular Doppler probe placed around the exposed and injured artery, which is a well-established and reliable method for assessing the antithrombotic properties of therapeutic agents [23].

In agreement with previous ex vivo results [9], the antithrombotic effect of *Vaa*-snaclec-3/2 in mice is due to the induction of thrombocytopenia. As demonstrated, *Vaa*-snaclec-3/2 induced a dose-dependent reduction in the platelet count in the peripheral venous blood of mice and caused profound thrombocytopenia at 20 µg/kg and higher doses. At a dose of 20 µg/kg, *Vaa*-snaclec-3/2 induced a decrease in the platelet count to approximately 10% of the normal value, which was protective for all FeCl_3_-treated animals (ED_100_). Our findings are consistent with those of a mouse model of thrombocytopenia, in which a decrease in the platelet count to 10% of the initial value strongly protected against occlusive thrombus formation [20].

The most likely mechanism underlying *Vaa*-snaclec-3/2-induced thrombocytopenia is platelet agglutination through the binding of *Vaa*-snaclec-3/2 to the GPIb receptor on these cells [17]. In an ex vivo study, platelet agglutinates were observed in human peripheral blood smears after exposure to *Vaa*-snaclec-3/2 [17]. In this *in vivo* study, no platelet agglutinates or clusters were detected in the peripheral blood smears; however, small platelet clusters were detected in the small pulmonary veins (Figure 5). The platelets within these clusters exhibited well-defined outlines and did not attach to the vein wall, supporting the conclusion that they are agglutinated rather than aggregated. No platelet clusters or thrombi were observed in the other organs examined, namely, the heart, kidney, liver, and spleen. Given that the lungs of the *Vaa*-snaclec-3/2-treated mice were slightly, although not significantly, heavier than those of the control mice, we suggest that the lungs are the primary site of platelet sequestration during *Vaa*-snaclec-3/2-induced thrombocytopenia in mice. The possibility of platelet sequestration in the mouse spleen is unlikely because it was previously demonstrated in guinea pigs that snaclec-induced thrombocytopenia occurred in splenectomized animals [24]. However, in patients with thrombocytopenia following *Vaa* envenomation, no signs of pulmonary involvement have been observed [17]. This finding may be due to the small size of the platelet agglutinates, which is insufficient to induce clinically significant vascular congestion, and because of the transient nature of thrombocytopenia [6,23].

In this study, thrombocytopenia in mice was partly reversed (*p* = 0.016) by F(ab’)_2_ fragments of IgGs raised against whole *Vaa* venom. This finding is consistent with previous findings in *Vaa*-envenomed human patients, where Fab [6] and F(ab’)_2_ [23] IgG fragments rapidly restored the platelet count [4]. The reversibility of thrombocytopenia further suggests that platelet clusters arise from platelet agglutination rather than aggregation, as the latter requires platelet activation. This interpretation is supported by both ex vivo and clinical studies [17].

In this study, no schistocytes were observed in peripheral blood smears, and histological examination revealed no evidence of hemoglobin release into the interstitial space of the examined organs. This finding indicates the absence of intravascular hemolysis or microangiopathic processes in the blood of the *Vaa*-snaclec-3/2-exposed mice. Our findings indicate that a reduced platelet count is not associated with thrombotic microangiopathy, of which schistocytes are a hallmark. Similarly, no hemolysis was observed in patients envenomed by the venom of *Vaa* [17].

Bleeding is the most concerning consequence of thrombocytopenia. However, it should be emphasized that no spontaneous hemorrhage was detected macro- or microscopically in this study. Moreover, histological examination revealed no evidence of acute bleeding. Capillaries contained only erythrocytes. Some of them were stained brown and clumped, which is most likely the consequence of oxidative hemoglobin modifications rather than iron accumulation [20]. *Vaa*-snaclec-3/2 is not an enzyme; thus, it lacks the capacity to directly induce hemoglobin oxidation. The observed brown staining of erythrocytes in the lungs, kidneys, or spleen is more likely due to oxidative modifications of hemoglobin (e.g., auto-oxidation to methemoglobin or other oxidized forms) within intact red blood cells, rather than hemolysis. This interpretation is further supported by the unchanged hemoglobin and hematocrit values (Figure 4).

Hemoglobin and hematocrit levels remained stable throughout the experiment, indicating the absence of clinically significant spontaneous bleeding, even in mice treated with the highest dose of *Vaa*-snaclec-3/2 (50 µg/kg), in which the platelet count decreased to just 2% of the normal value. This observation is consistent with previous *in vivo* studies showing that severe thrombocytopenia (up to a 97.5% reduction in platelet count) did not induce spontaneous bleeding in mice [20]. In this study, a tail bleeding time assay revealed that hemostasis in mice injected with 20 µg/kg *Vaa*-snaclec-3/2, the dose at which FeCl_3_-induced carotid artery thrombosis was completely prevented, was not compromised. Although significantly prolonged, the tail bleeding time remained within the reference range [19]. We may thus conclude that the *Vaa*-snaclec-3/2-induced increase in bleeding time is clinically insignificant, in line with clinical cases of *Vaa*-envenomed patients who suffer severe thrombocytopenia but have no clinical or laboratory signs of bleeding [3,16].

Clinical and ex vivo studies, along with the *in vivo* findings of this study, have revealed that *Vaa*-snaclec-3/2 induces rapid, intervention-driven fluctuations in the platelet count without activating platelets, impairing their function, or causing bleeding while effectively preventing thrombus formation after vessel injury [4,17]. Our findings demonstrate that the platelet count can be therapeutically modulated by *Vaa*-snaclec-3/2 without impairing platelet function, which represents a significant advance in antithrombotic research. Such an approach has great potential to drive the development of novel treatment strategies and drugs for interventional cardiology and angiology.

## 4. Conclusions

*In vivo*, *Vaa*-snaclec-3/2 induces profound thrombocytopenia while protecting mice from carotid artery occlusion. Thrombocytopenia does not cause spontaneous bleeding but does result in slightly prolonged tail bleeding following vascular injury, although these times remain within the reference range. In mice that were treated with *Vaa*-snaclec-3/2, low platelet counts in peripheral venous blood likely occurred due to platelet agglutination and possible sequestration in the lungs. Our results confirm that *Vaa*-snaclec-3/2 is a promising antithrombotic drug lead with particular value for the transient prevention of thrombosis, for example, in interventional angiology and cardiology.

## 5. Materials and Methods

### 5.1. Materials

*Vaa*-snaclec-3/2 was purified from crude *Vaa* venom and then chemically and biologically characterized as previously described [17]. Antivenom VIPERFAV^®^ containing F(ab′)2 (100 mg/mL) was purchased from MicroPharm (Inresa Arzneimittel, GMbH, Freiburg, Germany) (lot T4A111V), and heparin and sterile saline were obtained from Braun (Melsungen AG, Melsungen, Germany).

### 5.2. Animals

Young adult male BALB/c mice aged 12–24 weeks, which were obtained from Inotiv (Udine, Italy), were allowed to acclimate to the Animal Breeding Facility of the Veterinary Faculty, University of Ljubljana, for 14 days. Five mice were housed in each cage (1284 L EUROSTANDARD TYPE II L, 365 × 207 × 140 mm, floor area 530 cm^2^; Tecniplast, Buguggiate, VA, Italy) in a room with a controlled temperature and relative humidity (T = 20–24 °C; RH = 40–60%) and a 12/12 light/dark cycle (lights on at 06:00 am). Autoclaved wood fibers (LIGNOCEL 3/4-S; J. Rettenmaier & Söhne GmbH + Co KG, Rosenberg, Germany) were used as bedding. Food (Teklad global 16% protein extruded, irradiated, 2916C; Inotiv, Udine, Italy) and tap water acidified with HCl (pH 3–4) were provided *ad libitum*. Male mice were used instead of female mice to avoid the effects of fluctuations in coagulation protein levels that occur during the estrous cycle [25]. All the procedures were performed in accordance with ethical standards and were approved by the Administration of the Republic of Slovenia for Food Safety, Veterinary Sector and Plant Protection (approval no. U34401-9/2021/4). The experiments were planned and conducted according to the 3R principle and the ARRIVE and PREPARE guidelines [26,27].

### 5.3. Assessment of Antithrombotic Effect in a Mouse Model of Carotid Artery Thrombosis

The antithrombotic effect of *Vaa*-snaclec-3/2 was investigated in a mouse model of FeCl_3_-induced carotid artery thrombosis. The animals were anesthetized via intraperitoneal (*i.p.*) administration of ketamine (Narketan, Gorzow, Poland; 100 μg/g body mass (BM)), acepromazine (Le Vet Beheer B. V, TV, Oudewater, The Netherlands; 2 μg/g BM) and xylazine (Chanazine, Chanelle Pharmaceuticals Ltd., Loughrea, Ireland; 10 μg/g BM) [28]. Afterward, the left carotid artery was surgically exposed. The MA0.5VB perivascular flow Doppler probe was placed around the artery and connected to the corresponding T420 perivascular flow meter (Transonic Europe B.V., Elsloo, The Netherlands). The amplified signal was digitized using a data acquisition system (Digidata 1550B; Molecular Devices, Sunnyvale, CA, USA) with a sampling rate of 500 Hz.

#### 5.3.1. Dose–Response Study

The first dose–response study was performed to determine the dose–response relationship with respect to the antithrombotic effect of *Vaa*-snaclec-3/2. A dose of 1, 5, 10, 20 or 50 μg of *Vaa*-snaclec-3/2 per kg of mouse BM was dissolved in sterile saline solution (0.9% (*m*/*v*) NaCl) and administered to the mice as a bolus via the tail vein. Heparin (200 IU/kg BM) was used as a positive control [17,29,30]. Five mice were given each dose (n = 5). The drug, heparin and sterile saline solutions (control with or without thrombus formation induction) were *i.v.* injected in a volume of 100 μL. Five minutes after tail vein injection, FeCl_3_-soaked filter paper (approximately 1 × 2 mm, soaked in 3.5% (*m*/*v*) FeCl_3_ solution) was placed on the wall of the carotid artery and maintained there for 3 min to induce thrombus formation. Blood flow was subsequently monitored for 30 min. Afterward, a 200 µL blood sample was collected from the retro-orbital sinus, and the platelet count was determined. The mice were then sacrificed, the masses of the whole body and selected organs were measured, and the relative organ mass was calculated as follows: organ mass/body mass × 100.

#### 5.3.2. Experiments with an Effective Dose 100 of *Vaa*-Snaclec-3/2

The minimal tested dose of *Vaa*-snaclec-3/2 that caused severe thrombocytopenia and completely prevented FeCl_3_-induced carotid artery thrombosis in all experimental animals (ED_100_) was 20 μg/kg. We decided to use this dose of *Vaa*-snaclec-3/2 in a set of experiments carried out to determine the effects of *Vaa*-snaclec-3/2 on thrombus formation, peripheral blood platelet count, hemostasis, and histology of selected mouse organs as follows.

##### Effect of *Vaa*-Snaclec-3/2 on Thrombus Formation

The antithrombotic effect of *Vaa*-snaclec-3/2 at the ED_100_ was investigated in a mouse model of FeCl_3_-induced carotid artery thrombosis, as described in Section 5.3. *Vaa*-snaclec-3/2 was dissolved in sterile saline solution (0.9% (*m*/*v*) NaCl), and a dose of 20 μg/kg in 100 μL of saline solution was administered to the mice via the tail vein. The control animals (n = 8) received 100 µL of sterile saline solution (0.9% (*m*/*v*) NaCl) each. Heparin (200 IU/kg) was used as a positive control and was administered to mice (n = 8) in a volume of 100 μL. After 30 min, at the end of the blood flow measurement, a blood sample (200 µL) was taken from the retro-orbital sinus to determine the platelet count. The mice were then sacrificed, the masses of the whole body and selected organs were recorded, and the relative organ mass was calculated as follows: organ mass/body mass × 100.

##### Effect of *Vaa*-Snaclec-3/2 on Peripheral Blood Platelet Count

Blood samples (200 µL) were collected from the retro-orbital sinuses of mice into collection tubes (Sarstedt AG & Co. KG, Nümbrecht, Germany) using heparinized glass Pasteur pipettes (Glass Pasteur Pipettes, GMBH + CO, Wertheim, Germany) after the blood flow measurement was completed (refer to Section 5.3.1). Platelets were counted using an Advia 120 analyzer (Siemens, Munich, Germany) within 60 min of sample collection. Early assessment of platelet depletion induced by *Vaa*-snaclec-3/2 administration was accomplished 5 min after the mice were injected with *Vaa*-snaclec-3/2 (n = 3). These values were used to assess the kinetics of thrombocytopenia.

To assess the reversibility of *Vaa*-snaclec-3/2-induced thrombocytopenia, a 20 μg/kg dose of *Vaa*-snaclec-3/2 was injected *i.v.* into mice (n = 7). Ten minutes later, 50 µL of the VIPERFAV^®^ antivenom diluted to 100 µL with saline was injected *i.v.* into the animals at a final dose of 3 mg/kg. Blood samples were taken approximately 3 h after the antivenom was administered and the number of platelets was counted.

##### Effect of *Vaa*-Snaclec-3/2 on Hemostasis

The effect of *Vaa*-snaclec-3/2 on hemostasis was evaluated using the mouse tail bleeding time assay. The mice were anesthetized via *i.p.* injection of ketamine, acepromazine and xylazine. Afterward, 100 µL of sterile saline solution (0.9% (*m*/*v*) NaCl) was administered *i.v.* to the control group animals (n = 8). Twenty μg/kg of *Vaa*-snaclec-3/2 was administered *i.v.* to each mouse in the test group (n = 8). After 10 min, a 5-mm segment of the tip of the tail of each animal was cut with a sharp scalpel, and the tail stump was immediately immersed in sterile saline solution (37 °C) for 30 min of bleeding. The time from the initial exit of blood from the wound to the end of bleeding was recorded and observed for any postbleeding events. Afterward, a 200 µL blood sample was collected from the retro-orbital sinus, and the platelet count was determined. The mice were then sacrificed, the masses of the whole body and selected organs were recorded, and the relative organ mass was calculated as follows: organ mass/body mass × 100.

##### Effects of *Vaa*-Snaclec-3/2 on Tissues

After the blood flow measurement was complete (refer to Section 5.3.1 and Effect of *Vaa*-Snaclec-3/2 on Thrombus Formation), the mice were sacrificed, and the selected organs were extracted. The relative organ mass was calculated using the following formula: organ mass/body mass × 100. Liver, heart, lung, spleen and left kidney samples were collected and fixed in 10% buffered formalin (Shandon Formal-Fixx 10% neutral buffered formalin; Thermo Fisher Scientific, Basingstoke, UK), dehydrated and embedded in paraffin. Histological sections (5 µm) were then cut, stained with hematoxylin and eosin (HE) and coverslipped (Gemini AS slide stainer and coverslips ClearVue; Thermo Fisher Scientific, UK). Then, histological images were captured and examined with a Nikon Eclipse Ni-UM light microscope equipped with a DS-Fi1 camera and NIS-Elements imaging software, NIS Elements BR 4.6 (Nikon Instruments Europe B.V., Badhoevedorp, The Netherlands). Representative images were created using Adobe Creative Cloud (Adobe Inc., San Jose, CA, USA).

### 5.4. Statistical Analysis

The data were statistically analyzed using SigmaPlot for Windows version 12.5 (Systat Software Inc., San Jose, CA, USA). The normality of the data distribution was assessed using the Shapiro–Wilk test, followed by a test for the equality of variance. Normally distributed variables (platelet count and relative organ masses; *p* > 0.05) are presented as the mean ± S.E.M., whereas nonnormally distributed variables (tail bleeding time; *p* < 0.05) are presented as the median (IQR). Changes in relative organ mass were analyzed with an independent *t* test. Platelet counts were compared using one-way analysis of variance (ANOVA) followed by the Holm–Sidak post hoc test for multiple comparisons. For bleeding time, when the assumption of equal variance was not met, comparisons between groups were performed using the nonparametric Mann–Whitney U test. Box-and-whisker plots were used to display the results. A *p* value  ≤  0.05 was considered to indicate statistical significance.

## Figures and Tables

**Figure 1 toxins-17-00523-f001:**
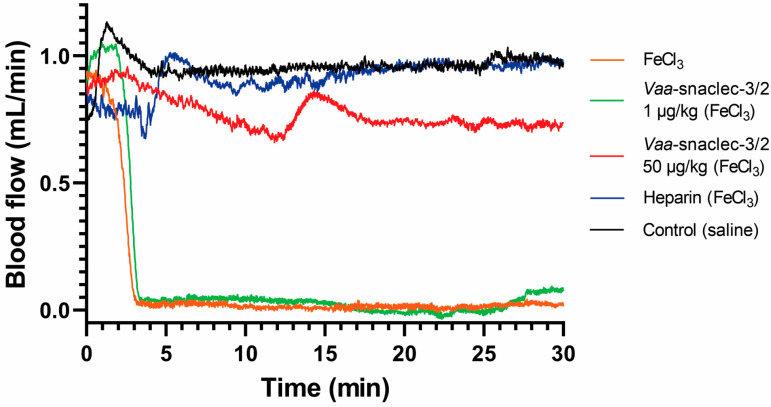
Effects of *Vaa*-snaclec-3/2 on mouse carotid artery blood flow. Representative recordings from control mice that were injected with 0.9% sterile saline solution (black line), mice injected with 0.9% sterile saline solution and with the carotid artery exposed to FeCl_3_ (orange line), mice injected with heparin and treated with FeCl_3_ (blue line), and mice injected with *Vaa*-snaclec-3/2 at doses of 1 μg/kg (green line) and 50 μg/kg (red line) and treated with FeCl_3_.

**Figure 2 toxins-17-00523-f002:**
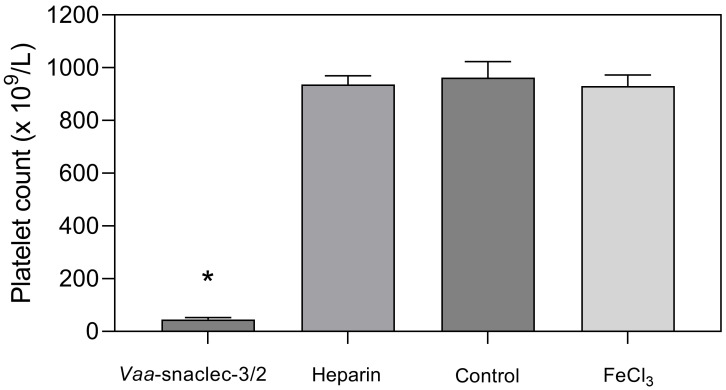
Platelet count values (mean ± S.E.M.) in the peripheral venous blood of mice 30 min after the induction of thrombus formation in the carotid artery with FeCl_3_. Animals (groups of 8) were first intravenously injected with *Vaa*-snaclec-3/2 (20 μg/kg), heparin, or sterile saline solution (control), after which their exposed arteries were externally treated with FeCl_3_ to initiate thrombus formation. The result designated as FeCl_3_ represents the platelet count in mice that did not receive any intravenous application. *Vaa*-snaclec-3/2 significantly reduced the platelet count; * *p* ≤ 0.05 versus control (*p* = 0.001). The platelet count data were first tested for normality (Shapiro–Wilk test). One-way analysis of variance (ANOVA) followed by the Holm–Sidak test was used for multiple comparisons.

**Figure 3 toxins-17-00523-f003:**
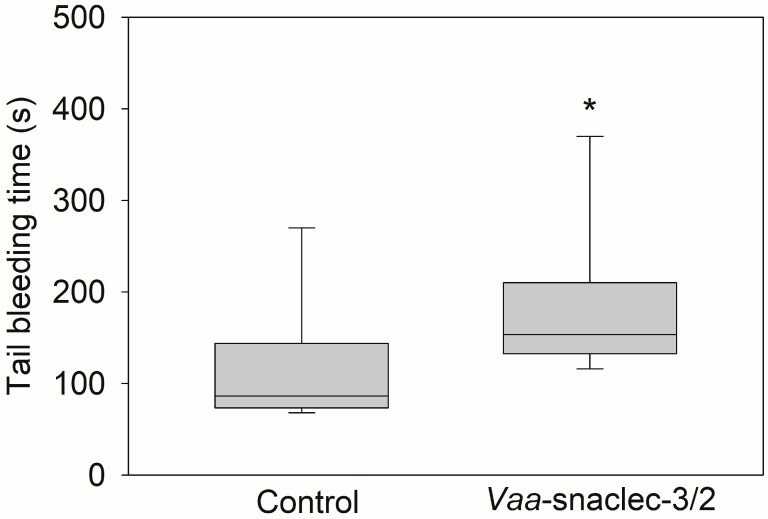
Effects of *Vaa*-snaclec-3/2 on mouse tail bleeding time. The mice were treated intravenously with 20 μg/kg *Vaa*-snaclec-3/2. The tip of the mouse tail was cut, and the bleeding time was recorded for 30 min as described in the Section 5. *Vaa*-snaclec-3/2 significantly increased the tail bleeding time; * *p* ≤ 0.05 versus control (Mann–Whitney rank sum test; *p* = 0.028). Boxplots show the median values of the data obtained from 8 mice per group and their interquartile ranges (IQRs).

**Figure 4 toxins-17-00523-f004:**
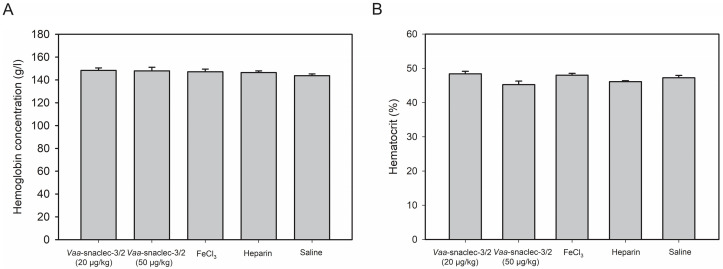
Hemoglobin concentration (**A**) and hematocrit values (**B**) in the peripheral venous blood of mice, measured 30 min after the induction of thrombus formation in the carotid artery with FeCl_3_. Animals were first intravenously injected with *Vaa*-snaclec-3/2 (20 or 50 μg/kg), heparin, or sterile saline solution (control), after which their exposed arteries were externally treated with FeCl_3_ to initiate thrombus formation. In one control group, exposed carotid arteries were externally treated with FeCl_3_ alone. *Vaa*-snaclec-3/2 do not significantly alter hemoglobin concentration or hematocrit values (*p* > 0.05) in any of experimental group. Data were first tested for normality using the Shapiro–Wilk test. One-way analysis of variance (ANOVA), followed by the Holm–Sidak test, was used for multiple comparisons. Values are expressed as mean ± S.E.M. (n = 4–7 per group).

**Figure 5 toxins-17-00523-f005:**
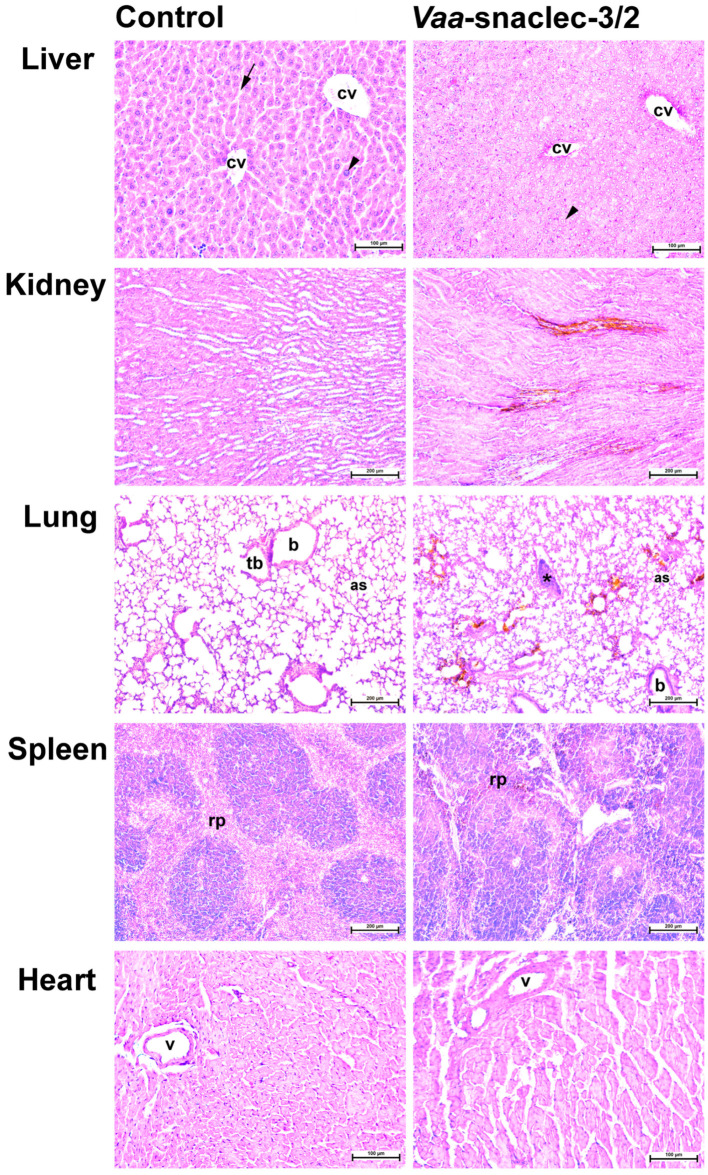
Representative photomicrographs of hematoxylin and eosin-stained liver, kidney, lung, spleen and heart sections of mice. The organ sections on the left represent controls. On the right are organ sections from a mouse treated *i.v.* with *Vaa*-snaclec-3/2 (20 μg/kg). Liver: arrowheads, karyomegaly (polyploidization); arrow, sinusoids; cv, central vein. Lung: as, alveolar sac; b, bronchiole; tb, terminal bronchiole; asterisk, platelet agglutinate; Spleen: rp, red pulp; v, coronary artery. Also note brown-stained erythrocytes in the kidney, lungs and spleen. Scale bars, 100 µm (liver, heart); 200 µm (kidney, lung, spleen).

**Table 1 toxins-17-00523-t001:** Relative organ masses of livers, hearts, lungs, spleens and left kidneys from the control group and the *Vaa*-snaclec-3/2 (20 µg/kg)-treated group.

	n	Mean	±S.E.M.	*p*
	Liver			
Control	8	4.325	0.105	*p* = 0.329
*Vaa*-snaclec-3/2	8	4.150	0.135	
	Heart			
Control	8	0.543	0.0176	*p* = 0.094
*Vaa*-snaclec-3/2	8	0.504	0.0142	
	Lungs			
Control	8	0.513	0.00715	*p* = 0.055
*Vaa*-snaclec-3/2	8	0.550	0.0155	
	Spleen			
Control	8	0.325	0.0129	*p* = 0.612
*Vaa*-snaclec-3/2	8	0.334	0.0129	
	Left kidney			
Control	8	0.786	0.0145	*p* = 0.612
*Vaa*-snaclec-3/2	8	0.751	0.0262	

## Data Availability

The original contributions presented in this study are included in the article. Further inquiries can be directed to the corresponding author(s).

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
