# Peer review of "Snake Venom C-Type Lectin-like Protein Vaa-Snaclec-3/2 Efficiently Prevents Carotid Artery Thrombosis in a Mouse Model Without Compromising Blood Coagulation"

_toxins, 2025, doi:10.3390/toxins17110523_

Round 1
Reviewer 1 Report
Comments and Suggestions for Authors
The authors in the current research article try to assess in vivo the antithrombotic potential of the C-type lectin-like protein from Vipera a. ammodytes viper venom using a mouse model of ferric chloride (FeCl3)-induced carotid artery thrombosis. While the list of methods and the experimental work design are appropriate and quite adequate, and the article's findings could be very interesting for professionals in this area of scientific interests and perspective for further investigation, it contains some confusing issues that need to be reformulated. It primarily concerns the lack of some important information and the overly brief discussion.
From the general remarks, I would point out the lack of more detailed information concerning the nature and function of snake venom C-type lectin-like proteins (which exhibit various activities and modes of action), as well as the known details about the structure and function of this particular one from the VAA venom.
There are no actual data presented for the experimental part where antivenom was used to reverse the effect of Vaa-snaclec-3/2; all described results are marked as “data not presented.” It would be important to see these data, possibly in a supplementary materials format.
Also, the discussion could be expanded further, and the proposed “unrelated oxidative changes that occurred in the tissue microenvironment” (page 9, lines 246-247) concerning the observed brown staining of erythrocytes should be explained.
The last sentence of the Discussion section is obviously a template remnant and should be removed (page 9, lines 269-270).
And, as an overall general remark, it needs to be clarified better from all this experimental work why the Vaa-snaclec-3/2 likely occurred due to possible sequestration in the lungs.
I think this work could be interesting for specialists, but only after some further revision and English editing.
Reviewer 2 Report
Comments and Suggestions for Authors
The authors have explored the possibility of Vaa-snaclec-3/2 isolated from the venom of towards the degradation of thrombus induced by FeCl3 in mice. The different doses of the compound was administered and the study results showed that 20 ug/kg body weight was effective as well as comparatively safe to be administered. However, severe decline in platelet count was observed after the administration of this compound, but there was no bleeding observed in the organs. The study is very extensive and the results are quite promising. However, I have a few comments:
- The authors have mentioned in the discussion that the haemoglobin level and haematocrit values did not vary. Where is the data?
- In the lungs there is agglomeration of platelets found. Can they cause embolism?
- Was there any behavioural changes observed in the animals after administration of the compound? Like anxiety or dull behaviour, induction of seizures or movement restrictions? Please report.
- The histopathological images of kidney and liver shows some visible differences, while nothing was reported in the manuscript. can you explain?
I recommend a major revision.
Round 2
Reviewer 1 Report
Comments and Suggestions for Authors
The presented corrections and additions are satisfactory, and the authors noticeably improved the article in accordance with the raised questions. This work can be accepted for publication in its present form.
Author Response
Thanks a lot for your efforts for improving the paper and for your final decision.
Reviewer 2 Report
Comments and Suggestions for Authors
The authors have made the changes as suggested by honorable reviewers. The article can be accepted for publication in its present form.
Author Response

(The authors gave the same response as above.)
